# Inflammatory Mechanisms in COVID-19 and Atherosclerosis: Current Pharmaceutical Perspectives

**DOI:** 10.3390/ijms22126607

**Published:** 2021-06-21

**Authors:** Marios Sagris, Panagiotis Theofilis, Alexios S. Antonopoulos, Costas Tsioufis, Evangelos Oikonomou, Charalambos Antoniades, Filippo Crea, Juan Carlos Kaski, Dimitris Tousoulis

**Affiliations:** 11st Cardiology Department, ‘Hippokration’ General Hospital, School of Medicine, National and Kapodistrian University of Athens, 11527 Athens, Greece; masagris1919@gmail.com (M.S.); panos.theofilis@hotmail.com (P.T.); alexios.antonopoulos@cardiov.ox.ac.uk (A.S.A.); ktsioufis@gmail.com (C.T.); boikono@gmail.com (E.O.); 2Radcliffe Department of Medicine, Division of Cardiovascular Medicine, University of Oxford, Oxford OX3 9DU, UK; antoniad@well.ox.ac.uk; 3Oxford Centre of Research Excellence, British Heart Foundation, Oxford OX3 9DU, UK; 4Oxford Biomedical Research Centre, National Institute of Health Research, Oxford OX3 9DU, UK; 5Department of Cardiovascular and Thoracic Sciences, Catholic University, 00168 Rome, Italy; filippo.crea@unicatt.it; 6Molecular and Clinical Sciences Research Institute, St George’s University of London, London SW17 0RE, UK; jkaski@sgul.ac.uk

**Keywords:** COVID-19, SARS-CoV-2, atherosclerosis, inflammation, anti-inflammatory treatment

## Abstract

Coronavirus disease 2019 (COVID-19) caused by the severe acute respiratory syndrome coronavirus 2 (SARS-CoV-2) has been associated with excess mortality worldwide. The cardiovascular system is the second most common target of SARS-CoV-2, which leads to severe complications, including acute myocardial injury, myocarditis, arrhythmias, and venous thromboembolism, as well as other major thrombotic events because of direct endothelial injury and an excessive systemic inflammatory response. This review focuses on the similarities and the differences of inflammatory pathways involved in COVID-19 and atherosclerosis. Anti-inflammatory agents and immunomodulators have recently been assessed, which may constitute rational treatments for the reduction of cardiovascular events in both COVID-19 and atherosclerotic heart disease.

## 1. Introduction

Initially affecting inhabitants of the city of Wuhan, Hubei Province, China in December 2019, the new severe acute respiratory syndrome coronavirus 2 (SARS-CoV-2), which led to coronavirus disease 2019 (COVID-19), manifested as a severe acute respiratory distress syndrome (ARDS) and was declared a pandemic on 30 January 2020, with high human-to-human contagiousness and mortality rate [1,2,3]. By January 2021, over 2,000,000 deaths and 90 million cases had been confirmed worldwide. Public health policymakers focused on public health measures to “flatten the curve”, while research efforts focused on assessing the efficacy of various pharmaceutical agents and the development of vaccines.

Close observation of thousands of severe COVID-19 cases has helped to gain insight into COVID-19 mortality rates and pathogenic mechanisms [4]. Clinical studies have demonstrated that COVID-19 mortality is predominantly related to thromboembolic disease and coagulation abnormalities [5], in which the so-called “cytokine storm” and systemic inflammation play an orchestrating role [5]. As inflammation plays an important pathogenic role in atherosclerotic cardiovascular disease in general and in ischemic heart disease in particular [5], we explored the similarities and differences regarding the inflammatory responses (cytokines, in particular) identified in atherogenesis and COVID-19 [6]. We also discuss the possible role of different treatment options that may affect the two conditions.

## 2. Inflammation in COVID-19

Coronaviruses (CoVs) are single-stranded RNA viruses that belong to the Coronaviridae family. The International Committee on Taxonomy of Viruses (ICTV) classifies the CoVs into four categories: α, β, γ, and δ. SARS-CoV-2 is the most recent coronavirus to infect humans. SARS-CoV, MERS-CoV, and SARS-CoV-2 are all viruses that cause extreme pneumonia. The SARS-CoV-2 genome is less than 30 kb in length and contains 14 open reading frames (ORFs) that encode non-structural proteins (NSPs) for viral replication and assembly processes; structural proteins such as spike (S), envelope (E), membrane/matrix (M), and nucleocapsid (N); and accessory proteins. COVID-19 is defined as an illness caused by the novel coronavirus. SARS-COV-2’s higher transmissibility, diverse clinical manifestations, and lower pathogenicity may be attributed to differences in biology and genome structure as compared to SARS-CoV and MERS-CoV.

SARS-CoV-2 enters human cells mainly by binding the angiotensin-converting enzyme 2 (ACE2), which is highly expressed by alveolar lung cells, vascular endothelium, cardiac myocytes, and other cells [7]. Patients suffering from cardiovascular or cerebrovascular comorbidities have a higher risk of infection and worse outcomes [8,9,10]. The virus spreads not only by inhalation of viral particles but also through contaminated surfaces, where it can live for 24–72 h depending on the type of surface. In most cases, the incubation time is shorter than 14 days, with the patient being contagious even though asymptomatic [7]. Commonly identified symptoms include fever, dry cough, dyspnea, chest pain, fatigue, and myalgia. Other less frequent symptoms are headache, dizziness, abdominal pain, diarrhea, nausea, and vomiting [11]. The virus decreases the Type-I Interferon (IFN) response and increases T cell apoptosis, as well as natural killers (NK) cell abnormalities. The direct attack and resulting immune system weakness may explain some of the extreme complications seen in COVID-19 patients, such as hypoxemia, ARDS, arrhythmias, trauma, acute myocardial injury, and acute kidney injury [2,7,12]. In the vast majority of COVID-19 patients, the typical chest computed tomography scan presents bilateral pulmonary parenchymal ground-glass and consolidative opacities, which is even worse in ICU-admitted patients with bilateral multiple lobular and subsegmental areas of consolidation [8,9,13,14]. Laboratory findings are not pathognomonic, with lymphopenia, prolonged prothrombin time, and elevated lactate dehydrogenase being the most prominent. Some patients with extreme bilateral pneumonia have elevated levels of aspartate aminotransferase, creatine kinase, creatinine, C-reactive protein (CRP), D-dimers, and ferritin [15].

## 3. The Cytokine Storm in COVID-19

The spectrum of symptoms ranges from asymptomatic infections to mild respiratory symptoms to the lethal form of COVID-19, which is associated with severe pneumonia, acute respiratory distress, and fatality. In the early stages of the disease, initial symptoms such as fever, cough, diarrhea, myalgia, or fatigue are present. COVID-19 patients may develop profound hypoxemia early in their disease course. However, overt respiratory failure at these early stages is unusual. Rarely, a minority of patients develop aggravating symptoms leading to multiorgan dysfunction and serious ARDS due to an intense inflammatory response and cytokine overproduction—the cytokine storm. Cytokines are small cell-signaling protein molecules, which may have autocrine or paracrine actions, facilitating intracellular crosstalk [16,17]. The cytokine family consists of more than 100 members, sub-categorized into several smaller clusters such as interleukins (ILs), INFs, colony-stimulating factors (CSFs), tumor necrosis factors (TNFs), and chemokines [6]. A cytokine storm resembling the hyperinflammatory state of COVID-19 has previously been recognized in several critically ill adults, namely those suffering from macrophage activation syndrome (MAS) or secondary hemophagocytic lymphohistiocytosis (sHLH). MAS/sHLH has historically been classified based on the cause of the disease and is categorized into primary (genetic) and secondary (non-genetic) types, and further subdivided into viral, autoimmune, or neoplasia-related [18]. The classic MAS/sHLH is characterized by fever, adenopathy, hepatosplenomegaly, anemia, other cytopenias, liver function abnormalities, and triggered hypercoagulation secondary to inflammation, followed by pronounced hypercytokinemia [12].

In immunodeficient patients with severe COVID-19 pneumonia, primary HLH may be the potential underlying cause. After entering respiratory epithelial cells, SARS-CoV-2 provokes the activation of Th1 cells to secrete pro-inflammatory cytokines. It is pictured by the failure of perforin, NK, and CD8+ cytotoxic T cells, which lead to cell lysis initiating apoptosis of virally infected cells. Additionally, IFN-γ, which is produced by a large number of widespread T cells, causes excessive macrophage activation [19]. In COVID-19 patients, the incidence of cardiovascular symptoms is high due to the systemic inflammatory response and immune system disorders during disease progression [20]. The exact mechanism of cardiac involvement in COVID-19 remains unclear. One potential mechanism is a direct myocardial involvement mediated via ACE2 or a direct viral effect on the heart muscle called myocarditis. Other proposed mechanisms of myocardial injury include cytokine activity and respiratory insufficiency, all of which may result in myocardial cell apoptosis through a gradual decrease in oxygen provided to the heart muscle [20]. In most immunocompetent patients with severe COVID-19 pneumonia, MAS/sHLH is suspected as the triggering cause of hyper-inflammation. The potential association between SARS-CoV-2 and sHLH relies on its key feature to bind Toll-like receptors (TLRs) and activate inflammasomes (caspases) releasing IL-1β [21]. In vitro cell studies indicate that in the early stages of SARS-CoV infection, respiratory epithelial cells, dendritic cells (DCs), and macrophages exhibit a delayed release of cytokines and chemokines [22]. The induction of low levels of antiviral factor IFNs and high levels of pro-inflammatory cytokines (IL-1β, IL-6, and TNF) as well as specific chemokines (C-C motif chemokine ligand (CCL)-2, CCL-3, and CCL-5) describe this hyperinflammatory state [23,24]. Therefore, the severity of COVID-19 has been strongly associated with the volume of hypercytokinemia, referring to interleukins IL-1β, IL-2, IL-6, IL-7, IL-10, TNF-alpha, granulocyte colony-stimulating factor (G-CSF), IFN-γ-induced protein 10 kDa/CXCL10, monocyte chemoattractant protein 1 (MCP-1), and macrophage inflammatory protein 1-α, both in serum and affected tissues [12,19]. After the secretion of the pro-inflammatory cytokines IL-1, IL-6, and G-CSF, low levels of IFN-αβ and IFN-γ induce inflammatory cell infiltration through mechanisms involving the Fas–Fas ligand (FasL) or the TRAIL–death receptor 5 (DR5) and trigger the apoptosis of airway and alveolar epithelial cells. Endothelial and epithelial cell apoptosis damages the pulmonary microvascular and alveolar epithelial cell barriers, resulting in vascular leakage, alveolar edema, and gradually, hypoxia [25,26]. Finally, the accumulated mononuclear macrophages engage in phagocytic activity on the debris of dead cells and tissues, secreting more pro-inflammatory cytokines (TNF, IL-6, IL-1β, and inducible nitric oxide synthase (NOS)) and chemoattractants (such as CCL-2, CCL-3, CCL-5, CCL-7, and IFN-γ-induced protein 10) [25]. The pro-inflammatory feed-forward loop of cytokines on innate immune cells results in a cytokine storm, coagulopathy, and acute respiratory distress syndrome [27] (Table 1).

## 4. Inflammation and Pro-Inflammatory Cytokines in Atherosclerosis

Atherosclerosis is characterized by the formation of plaques in the vessel wall. These develop through complex pathophysiological pathways that involve pro- and anti-inflammatory cytokines, secreted by vascular endothelial cells, leukocytes, platelets, and mast cells [6,16]. Endothelial injury, impaired lipid metabolism, and hemodynamic disruption, followed by flow-mediated inflammatory changes in the endothelium, are the main steps in plaque formation and atherosclerosis progression [8]. Inflammation begins when the endothelial cells become activated and secrete adhesion molecules (intercellular adhesion molecule-1 (ICAM-1), vascular adhesion molecule-1 (VCAM-1), E-selectin, and P-selectin) and other inflammatory factors while the smooth muscle cells secrete chemokines and chemoattractants (monocyte chemoattractant protein-1 (MCP-1)), which jointly draw monocytes, lymphocytes, mast cells, and neutrophils into the arterial wall, followed by their migration to the sub-endothelial space [6,34]. The effect of two pro-inflammatory cytokines, TNF-alpha and IL-1, which promote the expression of cytokines, adhesion molecules as well as the migration and mitogenesis of vascular smooth muscle and endothelial cells, is of particular interest [35]. TNF-alpha and IFN-γ have also been associated with the disruption of endothelial cell junctions, leading to leukocyte transmigration, vascular permeability, and matrix degradation, all of which facilitate atherosclerosis development [6,34,35,36].

With regards to human monocytes, three subsets exist owing to differences in functions [37]. CD14^+^CD16^++^ monocytes, similar to murine lymphocyte antigen 6 complex (Ly6C) low monocytes, patrol the vasculature and are involved in the early inflammatory response, while CD14^+^CD16^+^ monocytes exert both phagocytic and anti-inflammatory actions. Classical CD14^++^CD16^−^ monocytes, which are similar to murine Ly6C high monocytes, differentiate to macrophages and engulf a large number of chemically modified (by reactive oxygen species) low-density lipoprotein (LDL) particles known as oxidized LDL (oxLDL), which are prone to phagocytosis [6,38]. The accumulation of oxLDL in the macrophages transforms them into foam cells, contributing in this way to atherosclerotic plaques [39]. T cells, B cells, neutrophils, dendritic cells, and myeloid cell proliferation are types of resistant chemoattracted cells found in atherosclerotic lesions [40].

Selectins and platelet endothelial cell adhesion molecule (PECAM-1) assist the accumulation of leukocytes into the sub-endothelial space, inducing the inflammation and volume of the lipid core [41]. Fibrous tissue is added to form a fibrous cap over the lipid-rich necrotic cores and just under the endothelium at the blood interface. With aging, the persistent unregulated action of proteolytic enzymes dissolves the fibrous tissue; the fibrous cap becomes thinner and weakens. This thin cap is prone to rupture, exposing the thrombogenic material of the internal arterial wall leading to thrombus formation. As far as the vulnerability of plaque is concerned, studies have focused on the collagenolytic action of matrix metalloproteinases and cysteine proteases in the plaque [42]. The expression of matrix metalloproteinases (MMPs) and their tissue inhibitors (TIMPs) increases the risk of plaque rupture, bleeding, and thrombosis [43,44]. Recent studies have shown that IL-1, as well as TNF-alpha and its superfamily of CD40/CD40L ligand, disrupt the fibrinolytic and anti-thrombotic actions triggering thrombotic events [45]. It has also been described that in later stages of atherosclerosis, cytokines such as TNF-alpha, INF-γ, IL-1, and IL-6 act differently, inducing smooth muscle cell apoptosis and matrix degradation, leading to plaque destabilization [46,47]. Similar effects have been noted following the increase of acute-phase proteins, the proliferation of foam cells, and the reduced secretion of NO. CRP has also been linked to a pro-thrombotic function, as it is associated with TF upregulation, decreased levels of prostaglandins (PGI2) and endothelial NOS, and the impairment of coagulation balance, as measured by the thromboxane A2/PGI2 ratio [48,49,50,51]. Finally, the triple activity of INF-γ has been found to destabilize the plaque (as it prevents smooth muscle differentiation), the procollagen-I gene expression, and the collagen crosslinking enzyme, lysyl oxidase [52,53,54].

## 5. Similarities in the Inflammatory Processes Operating in COVID-19 and Atherosclerosis

COVID-19 presents not only complications in the venous and arterial circulations, but also multiorgan dysfunction [22]. This excessive response has a lot in common with the systemic low-grade inflammation in atherosclerosis. Initially, the impaired vessel endothelium following exposure to cytokines leads to decreased vasodilation in both atherosclerosis and COVID-19, as there is a reduction in the secretion of NO in both situations. Moreover, an improper activation of the coagulation cascade has been observed in COVID-19 patients [55]. Coagulation activation and endothelial dysfunction could be the expression of the sustained inflammatory response associated with vascular inflammation. Interestingly, this endothelium-related prothrombotic state is more prevalent in the lungs than in the lower limbs, even lacking particular risk factors and a history of thromboembolism [55]. The American Heart Association (AHA) suggests that viral infection reduces the cohesion of atherosclerotic plaque and encourages the progression of atherosclerosis and coronary heart disease [55]. IL-1β and IL-6 have major roles in both diseases. IL-6 may be significantly elevated in patients with ARDS from SARS-CoV-2, inducing the secretion of acute phase reactants, although the magnitude of cytokine level is not specific for the disease [56]. Studies showed that the higher circulating cytokine levels are, the more severe manifestations of COVID-19 pneumonia are seen [56]. In the same way, IL-18 and IL-12 in combination with IL-1β induce IFN-γ secretion and promote Th cell and NK activity in both atherosclerosis and COVID-19 [57,58]. Additionally, there are similarities in the secretion of IFN-α, IFN-γ, and TGF-β, as well that of TNF-alpha [12]. It is hypothesized that the atherogenic cytokines found in the plasma of COVID-19 patients are regulated by the shedding of ACE2, the portal allowing SARS-CoV-2 cell entry, or after virus incursion intracellularly [12]. Finally, in COVID-19, the apoptosis of endothelial and epithelial cells in the pulmonary microvasculature and tissues causes the release of chemokines, mainly from the CXC family CXCL9 and CXCL10, MCP-1 (CCL2), CCL2, and CCL3, which can be found in atherosclerotic plaque formation, leading to monocyte/lymphocyte recruitment and infiltration into the subendothelium [23,59,60].

## 6. Inflammatory Responses: Differences between COVID-19 and Atherosclerosis

Atherosclerosis is a chronic disease affecting the medium-sized and large arteries, while COVID-19 was thought to be a lung-centric viral infection with direct detrimental effects on vessels and heart via several potential mechanisms [61]. Firstly, hypoxemia caused by lung damage causes a progressive reduction in circulating oxygen partial pressure and saturation. With myocardial cell injury, oxygen free radicals, lactic acid, and metabolite aggregation occur [61]. Hypoxemia, oxidative stress, and raised pulmonary pressure, as a result of pulmonary vascular thrombosis, lead to cardiac dysfunction and heart failure [62]. Tissue hypoxemia induces a systemic inflammatory response, which can lead to an inflammatory storm [62]. Clinical observations have shown that SARS-CoV-2 binding to ACE2 results in an excessive release of Angiotensin II through the Renin-Angiotensin-System (RAS), leading to excessive vasoconstriction. This burdens the myocardium and vascular system by raising heart loading, which gradually leads to left ventricular hypertrophy and high blood pressure [7,62]. In some other cases, patients with a heart attack seem to suffer from a marked inflammation of the heart muscle—myocarditis. It remains unclear if myocarditis is caused by a direct effect of the virus on the heart muscle or by an overactive immune reaction to the virus [63]. Antiviral drugs and high catecholamine secretion (triggered by the inflammatory stress) have been proposed as independent burdening causes [63].

There are dissimilarities not only in the involved mechanisms but also in the secretion time and the nature of the inflammatory cytokines, as well as in their signaling pathways. In mild to moderate COVID-19, Type I IFN mostly contributes to a rapid reduction of viral load, preventing T-cell depletion and hypercytokinemia [64]. In severe COVID-19, a slow or insufficient antiviral response results in increased lung cytokine/chemokine levels and a compromised virus-specific T-cell response, demonstrating that cytokines rise exponentially in the first few days [65]. During the acute inflammatory response, a rapid (within 30 min) increase in TNF-alpha and IL-1β levels and a subsequent rise in IL-6 have been revealed. The high levels of IL-6 last for longer, while TNF-alpha and IL-1β levels rapidly decrease (within 24–48 h) [66]. Moreover, INF-α and INF-γ are intracellularly secreted soon after the viral invasion, as they are located in the arterial intima in the later stages of atherosclerotic plaque formation [19]. Despite the lack of sufficient data focusing on COVID-19 pathophysiology, clinical studies have reported high levels of interleukins IL-33, IL-21, and IL-17 as well as different chemokines of the CXC family [67].

During the progression of atherogenesis, there are not only atherogenic but also anti-atherogenic cytokines that have not been described in COVID-19 patients [68,69]. The central atheroprotective cytokines are TGF-β and several interleukins (IL-5, IL-10, IL-13, IL-19, IL-27, IL-33, IL-35, IL-37), which boost the T helper cells’ activity and downregulate TNF-alpha production [68,70]. Furthermore, pro-inflammatory mediators such as Oncostatin M and Cyclophilin A have been linked to endothelial cell expression of inflammatory cytokines and adhesion molecules [71,72] Finally, in atherosclerosis, macrophages, epithelial cells, and smooth muscle cells release bone-related cytokines such as Osteopontin and Osteoprotegerin, which stimulate matrix turnover, cell proliferation, and inflammation in plaques. Although a recent study on critically ill COVID-19 patients showed that serum osteopontin levels can be used to predict the severity of illness, no other trials have associated bone-related cytokines with the viral infection [73,74] (Table 2) (Figure 1).

## 7. Therapeutic Implications of Inflammation in COVID 19 and Atherosclerosis

A strong association between SARS-CoV-2 and RAS, which regulates renal, cardiovascular, and immune utilities, has been described. This relation is supported by the capacity of SARS-CoV-2 to adhere via its spike glycoprotein S to the metallopeptidase ACE2 receptor, which is expressed on the surface of the epithelial cells of vessels such as the lung, kidney, intestine, and heart, as well as on cerebral neurons and immune monocytes/macrophages [12]. Anti-inflammatory medications could constitute possible therapeutic options in the management of both diseases.

### 7.1. Atherosclerosis

Statins’ pleiotropic properties are well-known, and they have become the gold standard therapy for cardiovascular risk reduction and prevention [75]. Statins inhibit the critical step of cholesterol synthesis in which **3**-hydroxy-**3**-methylglutaryl coenzyme A (HMGC) is transformed to mevalonate by the enzyme HMGC reductase [75]. They have been proven to reduce serum cholesterol along with a significant reduction in morbidity and mortality from cardiovascular disease [75]. Statins have anti-inflammatory properties since the mevalonate pathway affects endothelial activity, inflammatory response, and coagulation [76]. Moreover, they stabilize the atherosclerotic plaque with thickened fibrous caps and macrocalcification via the reduction in lipid content [76]. The JUPITER study, the largest to date, found a notable decrease in major adverse cardiovascular events (MACE) associated with lower high sensitive CRP (hsCRP) levels in the rosuvastatin group versus the placebo group [77].

Antiplatelet agents are the pillar of CAD treatment, with acetylsalicylic acid being the most researched compound in this group. Aspirin enhances cell apoptosis by decreasing the cell proliferation rate of vascular smooth muscle cells to inhibit atherosclerosis progression by down-regulating the expression of Nf-κΒ subunit 1 (NF-κB1) and its targets, potentially providing us with more preventive techniques for disease control [78]. Atherosclerotic animal trials in LDL receptor-deficient mice recorded decreased levels of pro-inflammatory cytokines, eventually leading to diminished NF-κB activation, and improved atherosclerotic plaque stabilization following acetylsalicylic acid therapy [79].

Colchicine has been investigated as an antagonist of the (NOD)-like receptor protein 3 (NLRP3) inflammasome and as a result of IL-1 activity. NLRP3 inflammasome is a protein complex with an important role in vascular inflammatory progression [80]. The COLCOT study showed that starting colchicine shortly after myocardial infarction reduced the risk of cardiovascular mortality, resuscitated cardiac arrest, MI, stroke, or immediate hospitalization for angina involving coronary revascularization while a significant drop of hsCRP was observed [81,82].

Among immunomodulators, anakinra, an IL-1 receptor antagonist, has been found to downregulate CRP and IL-6 in patients with myocardial infarction, with further evaluation needed [6,83]. A cohort trial of patients with rheumatoid arthritis, in a 30-day follow-up, showed that the administration of anakinra reduced IL-6 and CRP levels, thus improving left ventricular function [84]. The CANTOS study also showed that canakinumab, a monoclonal antibody inhibitor of IL-1β, decreased the risk of cardiovascular events by lowering systemic inflammation in high cardiovascular risk patients. About 10,000 patients with a history of myocardial infarction and elevated hsCRP levels were included in the trial. Patients were divided into two groups: canakinumab and placebo, with a median follow-up of 3.7 years. The reduction in hsCRP and IL-6 was dose-dependent, while canakinumab at 150–300 mg per three months was observed to reduce MACE by approximately 15% [6,85]. Similarly, tocilizumab has been shown to increase endothelial activity and decrease aortic stiffness in atherosclerotic patients [81,82,83], with lower levels of CRP and fibrinogen observed [86,87,88]. A 17 mg/dL rise in low-density lipoprotein (LDL) has been noted, while another analysis reported a 22% increase in LDL and total cholesterol as well as a 48% increase in fasting triglycerides in these patients [89]. These results point to the need for further drug evaluation and more cautious patient selection. Another target is TNF, whose inhibitors are available and well-evaluated drugs without severe adverse events. The ENTRACTE study showed that the anti-TNF-alpha antagonists, etanercept and adalimumab, yield beneficial effects in atherosclerosis, blunting the progression of the subclinical disease via a decrease in ICAM and asymmetric dimethylarginine levels [90,91]. Finally, in patients with rheumatological conditions, TNF-alpha inhibitors showed a significant reduction of MACE [92,93]. These findings suggest that immunomodulators can play an important role in the therapeutic armamentarium of atherosclerotic disease.

### 7.2. COVID-19

Corticosteroids are potent cytokine inhibitors, acting mainly by inhibiting the NF-κB transcription factor [3]. Randomized controlled trials revealed that dexamethasone administration helps SpO_2_ levels to reach more than 90% in each case. Moreover, it decreases hospitalization, intubation, and the number of patients who do not need mechanical ventilation [3,94]. The mortality rates during hospitalization and one month after admission were also improved [3,94,95]. In a recent meta-analysis that included 1703 critically ill patients with COVID-19, the administration of corticosteroids was associated with lower all-cause mortality at 28 days. There was no indication of an elevated risk of adverse effects, while survival rates were similar to those in the dexamethasone and hydrocortisone groups. A higher dose of corticosteroids did not provide greater benefits than a lower dose [3].

Targeting the IL-1 family, anakinra has been found to downregulate CRP and IL-6 and is associated with significant survival improvement in severe COVID-19 [6,83]. In a cohort study of patients with COVID-19 and ARDS receiving anakinra, patients survived with no need for non-invasive ventilation outside of the ICU, while treatment with a high-dose was safe and associated with clinical improvement in 72% of the patients [96]. An observational study found that using anakinra on top of methylprednisolone reduced mortality in patients with hyperinflammation, respiratory dysfunction, or mechanical ventilation [97].

From among the IL-6 options, tocilizumab—formally included in the National Health Commission of China’s COVID-19 diagnosis and treatment program, and recently, in Infectious Diseases Society of America—is an IL-6 receptor antagonist that inhibits signal transduction by binding sIL-6R and mIL-6R and can be used in patients with bilateral pneumonia from SARS-CoV-2 [86]. Clinical data showed that the clinical image, hypoxemia, and opacity changes on computed tomography were ameliorated immediately after the treatment with tocilizumab in the vast majority of patients. The treatment was linked with improved overall recovery but a prolonged hospital stay due to metabolic, respiratory, and infectious adverse events [98]. Common side effects of tocilizumab include upper respiratory tract infections, headache, hepatotoxicity, hypertension, and infusion-related reactions [98]. Major adverse effects include hematologic effects, infections, gastrointestinal perforations, and hypersensitivity reactions [86,87,88]. Sarilumab, another anti-human IL-6 receptor monoclonal antibody previously used in the treatment of rheumatoid arthritis, showed positive effects in cases of COVID-19 with severe manifestations [99].

Anti-TNF-alpha antagonists etanercept and adalimumab could be also administrated in moderate COVID-19 with pneumonia in order to downregulate IL-1 and IL-6 levels as potential endpoints [90,91]. Finally, colchicine is a long-established drug that inhibits IL-1β by blocking pyrin and NLRP3 inflammasome activation. The possible advantage is based on limiting the occurrence of myocardial necrosis and pneumonia in cases of mild to serious COVID-19 [100,101,102]. It was reported that patients taking colchicine were five times more likely to be released within 28 days of admission, with a lower mortality rate (9% vs. 33% for those who did not take the drug) [102]. The GRECCO study showed that colchicine reduced the time between symptom onset and clinical improvement. However, there were no significant differences in levels of high-sensitivity troponin or CRP [101].

Regarding the commonly prescribed agents in atherosclerotic cardiovascular diseases with proven anti-inflammatory effects, their use has been associated with favorable outcomes in observational studies. Starting with statins, their use was associated with significantly lower mortality according to data from a single-center Polish registry and two recently completed retrospective studies using propensity score matching; they have also been associated with reduced intensive care unit admissions and other COVID-19 complications. With regard to aspirin, newly reported studies highlight its association with reduced mortality and serious complications like mechanical ventilation or intensive care unit admission, with a recently completed systematic review and meta-analysis confirming those findings despite a low certainty of evidence.

Randomized controlled trials are currently underway to test the efficacy of the above-mentioned drugs in atherosclerosis and especially in COVID-19 patients, with very hopeful results until now (Table 3).

## 8. Conclusions

Endothelial dysfunction, hyperinflammation, and coagulopathy contribute to disease severity and death in patients infected with SARS-CoV-2, while also being prevalent features of atherosclerosis. The presence of a cytokine storm in patients with COVID-19 causes ARDS or multiorgan dysfunction, including an increased risk of plaque rupture and direct myocardial injury (i.e., myocarditis), leading to high mortality rates. An optimal regulation of the cytokine storm in the early stages of the disease can contribute to treatment effectiveness and reduce the risk of cardiovascular complications, which are the leading cause of death in these patients. In atherosclerosis, anti-inflammatory treatment could also modify the progression of the disease, as shown in recently completed trials that demonstrated the efficacy of IL-1β inhibition on the reduction of cardiovascular risk. Thus, the results of randomized clinical trials and ongoing observational biospecimen studies concerning anti-inflammatory treatment may provide additional clues for the mitigation of inflammation and the further improvement of outcomes in patients with COVID-19 and atherosclerotic disease. Intriguingly, pharmacological agents frequently used in atherosclerotic conditions, such as statins and aspirin, appear to lower the incidence of serious COVID-19 complications and mortality rates—findings which deserve validation in randomized settings. Last but not least, despite the pivotal role of endothelial dysfunction in the early stages of both diseases, direct treatment remains suboptimal, highlighting the need for further research in this direction.

## Figures and Tables

**Figure 1 ijms-22-06607-f001:**
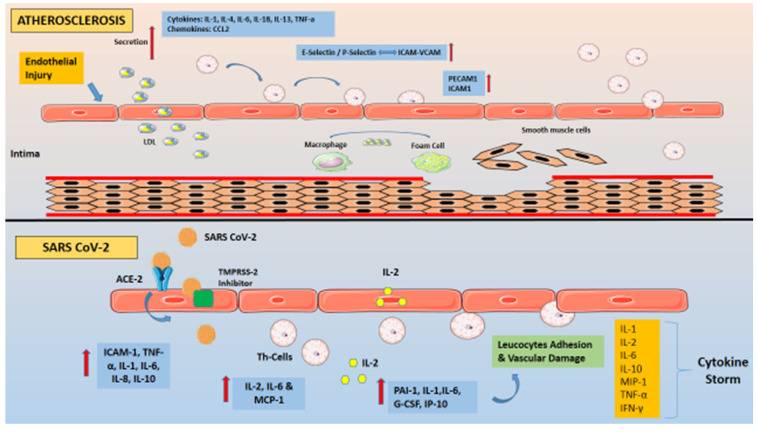
(i) Atherosclerosis: Involves the activation of the endothelium, which upregulates inflammatory signaling mechanisms via the release of inflammatory cytokines and chemokines. LDL particles accumulate in the tunica intima and are subjected to oxidative modification. Circulating monocytes bind to adhesion molecules expressed on the activated endothelial cell surface and migrate to the intimal layer. Subsequently, they transform to macrophages and express receptors that bind to oxidized LDL particles, ultimately turning into foam cells. T lymphocytes as well as vascular smooth muscle cells situated in the medial layer, also migrate to the intimal layer. (ii) COVID-19: “The role of endothelial cells in SARS-CoV-2 infection”. Severe acute respiratory syndrome coronavirus 2 (SARS-CoV-2) binds with Angiotensin-converting enzyme 2 (ACE2) on the cell membrane of the host cells. Cell invasion also depends on the presence of the protease Transmembrane protease serine 2 (TMPRSS-2) that can cleave the viral spike. The recombinant protein of human ACE2 fused with the Fc region of the human immunoglobulin IgG1 (rACE2-IgG1) binds with high affinity to the receptor-binding domain of SARS-CoV-2. In patients diagnosed with severe COVID-19, increased levels of pro-inflammatory cytokines, particularly the soluble interleukin 2-receptor (IL-2R) and interleukin-6 (IL-6), have been observed. Endothelial cells (ECs) express both the IL-6 receptor (IL-6R) and IL-2R on their surface. Soluble IL-2R (sIL-2R) is mostly secreted by activated T helper lymphocytes but might also be secreted by ECs. The binding of IL-6 and IL-2 on their receptors induces a capillary leak. Moreover, IL-6 signaling induces the secretion by ECs of more IL-6 and other cytokines. The continuous burdening of the endothelium leads to further secretion of inflammatory cytokines and the over-reaction of the immune system, the so-called “Cytokine Storm”. ΡAΙ-1 = Plasminogen activator inhibitor-1, TNF = Tumor Necrosis Factor, ICAM = Intercellular Adhesion Molecule 1, VCAM = Vascular cell adhesion protein 1, PECAM = Platelet endothelial cell adhesion molecule-1, MCP-1 = monocyte chemoattractant protein-1, G-CSF = Granulocyte colony-stimulating factor, IP-10 = Interferon gamma-induced protein 10, MIP-1 = Macrophage inflammatory protein-1, IFN = Interferon.

**Table 1 ijms-22-06607-t001:** Cytokines involved in COVID-19 and their prognostic data.

IL-/TNF- and IFN-Family Cytokines	
Factor	Prognostic Value
IL-1β	Elevated levels IL-1β have been associated with hypercoagulation, disseminated intravascular coagulation, and severe symptoms [28].
IL-2	Increases in IL-2 or its receptor IL-2R are directly proportional to the severity of the disease [13].
IL-4	IL-4 has negative effects on CD8+ memory T cells; elevated IL-4 levels are associated with cytokine storm and severe respiratory symptoms [29].
IL-6	Higher levels of IL-6 accelerates the inflammatory process, contributing to the cytokine storm and worsening the prognosis [18].
IL-12	NA
IL-17	Elevated IL-17 levels have been reported in patients with SARS-CoV-2 as part of the cytokine storm, and they are associated with viral load and disease severity [30].
IL-18	NA
IL-21	NA
IL-33	Higher IL-33 levels have been associated with lung fibrosis and skeletal muscle wasting [31].
TNF-alpha	TNF-alpha was one of the cytokines whose overproduction was related to a poor prognosis in patients with SARS-CoV-2, finding an inverse relationship between TNF-alpha levels and T cell counts [32].
TGF-β	NA
IFN-α	NA
IFN-γ	IFN-γ levels are associated with greater viral load and lung damage [33].
**Chemokines**	
CCL2/MCP-1	CCL2 levels were higher in patients with COVID-19 and even higher among those admitted to the Intensive Care Unit [9].
CCL3/MIP-1A	NA
CCL5	NA
CXCL9	NA
CXCL10/IP-10	IP-10 levels were found to be elevated in patients with COVID-19 and even higher in those who required Intensive Care Unit admission, suggesting their relationship with lung damage and disease severity [9].

IL- = Interleukin, TNF-alpha = Tumor Necrosis Factor-alpha, TGF-β = Transforming Growth Factor-β, IFN- = Interferon, CCL = C-C Motif Chemokine Ligand, CXCL = C-X-C Motif Chemokine Ligand, MCP-1 = Monocyte chemoattractant protein-1, MIP-1A = Macrophage inflammatory protein-1A, IP-10 = Interferon gamma-induced protein 10.

**Table 2 ijms-22-06607-t002:** Inflammation in Atherosclerosis and COVID-19—Similarities and Differences.

Similarities		
Factor	Atherosclerosis	COVID-19
NO^−1^	Decreased	Decreased
Coagulation Factors	Increased	Increased
IL-1β	Increased	Increased
IL-6	Increased	Increased
IL-12	Increased	Increased
IL-18	Increased	Increased
IFN-α	Increased	Increased
IFN-γ	Increased	Increased
TGF-β	Increased	Increased
TNF-alpha	Increased	Increased
CCL2	Increased	Increased
CCL3	Increased	Increased
CXCL9	Increased	Increased
CXCL10	Increased	Increased
C-Reactive Protein	Increased	Increased
**Differences**		
**Factor**	**Atherosclerosis**	**COVID-19**
Angiotensin II	NA	Increased
IL-3	Increased	NA
IL-8	Increased	NA
IL-15	Increased	NA
IL-17	NA	Increased
IL-21	NA	Increased
IL-33	NA	Increased
M-CSF	Increased	NA
CXCL8	NA	Increased
CXCL11	Increased	NA
CXCL16	Increased	NA
Oncostatin M	Increased	NA
Cyclophilin A	Increased	NA
Osteopontin	Increased	NA
Osteoprotogerin	Increased	NA
Ferritin	NA	Increased

**Table 3 ijms-22-06607-t003:** Beneficial anti-inflammatory therapeutic options.

Drugs	Action	COVID-19 Ongoing Trials	Atherosclerotic Disease Trials
**Corticosteroids**	Immunosuppression	NCT04273321	-
**Anakinra**	Monoclonal antibody against IL-1 Receptor	NCT04339712, Phase 2	Ikonomidis et al. [84]
**Emapalumab**	Monoclonal antibody against IL-1 Receptor	NCT04324021, Phase 2/3	-
**Canakinumab**	monoclonal antibody against IL-1-beta	NCT04362813, Phase 2	CANTOS Trial [102]
**Tocilizumab**	IL-6 Receptor Inhibitor	NCT04317092	Holte et al. [103]
**Sarilumab**	IL-6 Receptor Inhibitor	NCT04280588, Phase2	-
**Heparin**	Anticoagulant, anti-inflammatory, antiviral	NCT04345848, Phase 3	-
**Colchicine**	Inhibition of NLRP3 inflammasome	NCT04326790	COLCOT Trial [80]
**Adalimumab**	Anti-TNF-alpha antagonists	Case Series	ENTRACTE Trial [103]
**Etanercept**	Anti-TNF-alpha antagonists	Case Series	ENTRACTE Trial [103]
**Aspirin**	Inhibitor of the enzyme cyclooxygenase (COX)	NCT04363840	Coronary Microvascular Angina Trial (CorMicA) [103]

## Data Availability

Not applicable.

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
