# Peer review of "Inflammatory Mechanisms in COVID-19 and Atherosclerosis: Current Pharmaceutical Perspectives"

_ijms, 2021, doi:10.3390/ijms22126607_

Round 1

Reviewer 1 Report

Overall, this is a well written and well-organized paper. The background or literature summary is thorough. The authors presented a fairly comprehensive summary of current understanding of inflammatory responses toward Covid-19 and the long-term progression of atherosclerosis. They also highlighted the therapeutic interventions for both diseases and the potential overlap of their beneficial effects. However, there are also several deficiencies in this paper in addition to other minor issues. Some passages and sentences are not clear. Those are discussed in detail below.

Major Comments

Starting in section 2, the terms SARS-CoV2 and COVID-19 are used at different places without much qualifier. The paper moves from one to the other without much interruption. It seems to suggest that SARS-CoV2 and COVID-19 are very similar, almost interchangeable in terms of the inflammatory responses they cause in patients. It’s obvious that the paper is trying to focus on COVID-19 the virus behind the current pandemic. But it switches between SARS-CoV2 and COVID-19, that it is hard to keep track, or to know the difference between the 2.

What is SARS-CoV2? And how is it similar or different from COVID-19? How different are the disease mechanisms differ, immune or inflammatory responses, or pharmaceutical therapeutics? Some background information on SARS-CoV2 would be helpful. Plus SARS-CoV2 is behind us for now. So what have we learned from SARS-CoV2 and its connection to atherosclerosis? What information could we gain from SARS-CoV2 that would help our fight against COVID-19?

Section 3, first paragraph: there is discussion on childhood disease resulting in MAS/sHLH. It is not clear how this is connected to COVID-19 discussion. COVID-19 does not have a significant impact in children, and neither does atherosclerosis. Please clarify that passage.

Section 4 is very short, and doesn’t need to stand alone. Consider combining that with Section 5.

Section 5, line 158 to 160: sentence that starts “In the same way…” appears to be incomplete.

In the same passage, what is meant by CRP? That was never spelled out.

Section 6, line 175: sentence starting with “Initially, the impaired vessel…” what is meant by impaired vessel epithelium?   

Section 6, line 177: sentence starting with “Moreover, there is a reduction in the secretion of nitric oxide in both situations.” Is there a reference for this finding?

Section 6, line 178: sentence starting with “Thus, flow-mediated hemodynamics…” what is meant by this sentence? This sentence brings in the idea of hemodynamic control on the disease progression, which is fine. But it’s not clear how this sentence links to the rest of this section.

Section 6, line 182: sentence starting with “This is another vessel-related pathway…” what is the pathway they are referring to?

Section 7, last paragraph on page 6, starting on line 235: the discussion suddenly switched to bone-related cytokine, and is the not discussed anywhere else in the paper. How does this finding relate to atherosclerosis or COVID-19? It is not clear what the purpose of this paragraph is, or how it connects to the rest of the paper.  

Overall, better link or transition from paragraph to paragraph or section to section would help improve the paper. Right now, it seems like sections of the paper are fairly independent can stand alone. Some areas feel like they are written be different authors.   

Section 8, line 274 under i) Atherosclerosis: the first sentence: “COVID-19 inflammatory cytokines and chemokines have a lot in common with these which presented in the early and later stages of atherosclerotic plaque formation [58]”. What is meant by “these”? It’s not clear why this first sentence is here.  h

Section 8, bottom of page 9, line 370: what is RCTs? It was not discussed till this point. 

Finally, based on the findings discussed in this paper, particularly Section 8, one question that could be proposed is what is the COVID19 infection or recovery rate of people have already been taking drugs like statin or aspirin long term? If there is crosstalk in inflammatory responses or therapeutics between COVID-19 and atherosclerosis, it’s possible that people on medication to reduce or control atherosclerosis might be better prepared to combat COVID-19 infection. This question could be proposed and discussed in Conclusions.

In addition, Section 9 Conclusion could be expanded on more. After drawing comparisons between atherosclerosis and COVID-19 throughout the paper, what are some of the major overlaps between the two diseases and their therapeutics that could benefit current treatment of COVID-19, for example. What other proposals or recommendation for future studies that could be drawn based on the findings from Sections 1 to 8. This paper needs a stronger finish in the final section, to highlight all the major conclusions and propose new questions for future analysis.

Minor Comments

Section 3 line 72 to 73:

“A cytokine storm leading to macrophage activation syndrome (MAS) or secondary haemophagocytic lymphohistocytosis (sHLH) has previously been identified in children with systemic-onset juvenile inflammatory arthritis -Still illness- [15].”

What is “arthritis – Still illness –“?

Names such as TNF (Tumor Necrosis Factor) IFN (Interferon), NFkB1, are used throughout the paper. Please be consistent in how these names and their acronyms are introduced and used. The first time they appear, the full name can be spelled out with the acronym or short name in parenthesis. And then after that, only the short name is needed. In addition, consider adding a table of acronyms at the beginning or end of the paper to help remind people.

TNF-“alpha” or “a” are both used. Please go with -alpha through out the paper.

Section 8, line 280: should be proven instead of “proven” instead of “proved.”  

Author Response

Dear Reviewer,  

We are grateful for the feedback, which was very helpful in order to improve the quality of our manuscript. We have tried to address each comment sufficiently in a point-by-point response. Please see the attachment.

Marios Sagris

Reviewer 2 Report

Thank you for the paper. It is interesting to compare the inflammatory response in COVID -19 and atherosclerosis, particularly if there are current medications for CVD that could be used in the treatment of COVID 19.
The concept is worth exploring. However, I feel that the paper tried to cover too much scope and depth at the same time; this made the paper very hard to follow at times. There was an excessive amount of information. To be a good review, you need to do more than tell information, you need to tell a story with the information - bring out a new perspective on a topic. The story presented was not always clear. This may have arisen from the angle that was taken in the paper. While the title says ‘inflammatory mechanisms’, the paper rushes into cytokines stating a myriad of facts about how they work; this distracted from what they do, and left the key message unclear. For eg, in cytokines in atherosclerosis there is a section:
(line 136 and following) TNF-a causes the activation of myosin light chain kinase by increasing cytosolic Ca2+. Their contribution with the expression of Ras homolog gene family member A (RhoA) disrupt the endothelial cells junction [5,27,28]. In addition, INF-γ suppresses the formation of F-actin stress fibers by altering the cadherin-catenin complex in the vascular endothelium, which promotes the inflammation [29]. The activation of Janus kinase 1 (JAK-1) by high levels of IL- 6 has been shown to promote the development of atherosclerosis in experimental mice following injection. [30,31].
Is the key point that TNF-alpha and IFNγ injure the endothelium as seen by a disruption in the endothelial cell junctions?  For this review, unless these pathways (Rho and F-actin) are those that are discussed as possible targets for therapeutic target later in the paper, then do we need to know that the action involves an increase in Rho or suppressed the f-actin stress fibres?  Rather, how does this disruption contribute to atherosclerosis development?  Does it increase permeability to LDL or leukocytes?
Also, with the focus on cytokines, the result is that the progression of atherosclerosis is out of order. We are told about plaque destabilization, line 153, before growth of the lipid core, in line 164. Yet growth of the core (as well as thinning of the plaque cap) are what constitute an unstable plaque morphology.
Also, some information presented is incorrect, which questions the understanding of the field.  For eg:
(line 147) ‘several monocytes differentiate to macrophages and engulf a large number of low-density  lipoprotein (LDL) leading to oxidized LDL –oxLDL- [5,32]’.
In vitro studies tend to show that LDL is not easily taken up by macrophages, where as oxLDL is taken up more easily,  ie the oxidation occurs before uptake. Reference 5, which is cited, clearly makes that the point that macrophages engulf oxLDL.
In terms of inflammation in COVID-19, the heavy focus on the cytokine storm, without its chronological setting, makes it look like the cytokine storm is the upfront initial immune response in everyone, rather than a dysregulated response in the minority.
I didn’t find the figure very helpful. Leukocyte recruitment across the endothelium is only the surface of the inflammatory response. Plaque growth is a lot more complex than the accumulation of cells. The tables were very good, and presented the information very clearly. 

The comparison of the similarities and differences (section 6 and 7) read much better, but a clearer setting of the stages of the diseases in the earlier focus on each disease (sections 2 and 4) would have made them even easier to follow.   
I feel that if the authors want to look at inflammation, they need to step back and look more chronologically at the development of the two conditions and show how the inflammation can lead to poor outcomes.   However, the authors may like to consider a different story, such as looking at CVD medications and the indications of their possible use in COVID. The paper at the moment, is trying to be too broad and deep at the same time.
The senior authors should have provided more feedback to the junior authors before submission. 

Author Response

(The authors gave the same response as above.)

Round 2

Reviewer 1 Report

Overall, paper is acceptable for publication. Please do a final check of English working, grammar and spelling, and any other formatting issues. 

Author Response

Dear Reviewer,  

We are grateful for the feedback, which was very helpful in order to improve the quality of our manuscript. We have tried to address each comment sufficiently in a point-by-point response below:

Marios Sagris

Reviewer 2 Report

I thank the authors for the revised manuscript; this version is much easier to read. I particularly like that they have more clearly set the scene of both conditions while taking away unnecessary detail. The flow is much better for these changes.

There are still some issues that I feel needed addressing.

  1. The heading says inflammatory mechanisms in COVID-19 and atherosclerosis.

I feel that the paper is mainly addressing cytokines, the role of cells is only lightly touched on. As such, while the title is okay in this respect, when you discuss what the review covers: we explored the similarities and differences regarding the inflammatory response identified in atherogenesis and COVID 19, can you stress the cytokine angle. Line 44: ‘We explored the similarities and differences regarding the inflammatory response (in particular, cytokines) identified in atherogenesis and COVID 19’.

  1. Section 4. Inflammation and pro-inflammatory cytokines in atherosclerosis.

The section flows better than before, and the progression is now more chronological with it good to see the development of atherosclerosis covered from initiation through to advanced. However, it still needs some work.

  1. line 136: In the sentence ‘The effect of atherosclerosis is progressive leading to formation of the atherosclerotic plaque in vessels,’ isn’t clear as it essentially says ‘atherosclerosis leads to atherosclerosis’. Perhaps just simply state ‘Atherosclerosis is characterized by the formation of plaques in the vessel wall. These develop through complex….’
  2. line 138. In this line we find a definition of what cytokines are, yet ‘cytokine’ is mentioned back in line 38. It looks as if this section was written by a different author than the main text and the flow of how the sections go together with each other has not been addressed. If a definition of cytokines is desired, then this needs to go earlier in the paper.
  3. This section:

‘Inflammation begins when the endothelial cells become activated and secrete adhesion molecules, and the smooth muscle cells secrete chemokines and chemoattractants, which together draw monocytes, lymphocytes, mast cells, and neutrophils into the arterial wall. In early atherogenesis, blood circulating monocytes adhere to the endothelium and migrate to the sub-endothelial space [5]. The activation of endothelial cells leads to the secretion of monocyte chemoattractant protein-1 (MCP-1), IL-8, intercellular adhesion molecule-1 (ICAM-1), vascular adhesion molecule-1 (VCAM-1), E-selectin, P-selectin, and other inflammatory factors, with lymphocytes and monocytes infiltrating the arterial wall.

These sentences need to be polished, by being more succinct as there are 3 concepts that are essentially repeated.

  1. Line 161/2 needs a reference. In general, mouse monocytes are considered Ly6C hi and Ly6Clo, with Gr-1 being Ly6C and Ly6G together. The Ly6chi considered to be inflammatory and the Ly6Clo considered to patrol the endothelium, though can play an inflammatory role in disease states. I am not familiar with a division based on Ly6Chi and Gr-1, so please cite a reference. 
  2.  
  3. Conclusion

Line 416: “IL inhibition on reduction of cardiovascular risk’. Was ‘IL’ meant to read ‘IL-1ß’?

Author Response

(The authors gave the same response as above.)
